# Dynamics of the Queensland Fruit Fly Microbiome through the Transition from Nature to an Established Laboratory Colony

**DOI:** 10.3390/microorganisms10020291

**Published:** 2022-01-26

**Authors:** Rajib Majumder, Phillip W. Taylor, Toni A. Chapman

**Affiliations:** 1Applied BioSciences, Macquarie University, North Ryde, NSW 2109, Australia; phil.taylor@mq.edu.au (P.W.T.); toni.chapman@dpi.nsw.gov.au (T.A.C.); 2Biosecurity and Food Safety, NSW Department of Primary Industries, Elizabeth Macarthur Agricultural Institute (EMAI), Menangle, NSW 2567, Australia

**Keywords:** *Bactrocera tryoni*, Tephritidae, high-throughput Illumina sequencing, domestication, sterile insect technique, gel-based diet

## Abstract

The transition from nature to laboratory or mass rearing can impose significant physiological and evolutionary impact on insects. The Queensland fruit fly (also known as ‘Qfly’), *Bactrocera tryoni* (Froggatt) (Diptera: Tephritidae), is a serious economic pest that presents major challenges for horticulture industries in Australia. The sterile insect technique (SIT) is being developed to manage outbreaks in regions that remain free of Qfly and to suppress populations in regions where this species is endemic. The biology of Qfly is intimately connected to its microbiome. Therefore, changes in the microbiome that occur through domestication have implications for SIT. There are numerous studies of the microbiome in Qfly larvae and adults, but there is little information on how the microbiome changes as Qfly laboratory colonies are established. In this study, high-throughput Illumina sequencing was used to assess the Qfly microbiome in colonies reared from wild larvae, collected from fruit, for five generations, on a gel-based larval diet. Beta diversity analysis showed that the bacterial communities from Generation 5 (G5) clustered separately from earlier generations. At the genus level, bacterial communities were significantly different between the generations and mostly altered at G5. However, communities were found similar at phyla to family taxonomic levels. We observed high abundance of *Morganella* and *Burkholderia* at the genus level in the larval and pupal stages respectively at G5, but these were not detected in earlier generations. Overall, our findings demonstrate that the domestication process strongly affects the Qfly microbiome and prompts questions about the functional relationship between the Qfly and its microbiome, as well as implications for the performance of insects that have been domesticated and mass-reared for SIT programs.

## 1. Introduction

Insect colonies are commonly established and maintained under artificial rearing conditions for research or, on a larger scale, for release in pest management programs as natural enemies or, after sterilisation, to disrupt reproduction of pest populations (‘sterile insect technique’, SIT) [1,2,3,4]. The conditions of artificial rearing are very different from nature, exposing insect populations to genetic drift and significant selection pressures. Such selection pressures favour those types that are able to develop, survive and reproduce under artificial rearing conditions, leading to domestication [5,6]. Domestication can have profound effects on insect physiology and behaviour, improving performance under artificial rearing conditions, but often as a consequence reducing their performance if returned to field conditions [7,8,9]. In tephritid fruit flies, domestication has been reported to have significant influence on numerous life history traits, including development, stress tolerance and reproductive behaviour. Compared to wild type flies, domesticated fruit flies tend to mature at an earlier age and may have reduced sexual competitiveness, reduced compatibility with wild populations and reduced environmental tolerance [10,11,12,13,14]. The microbiome is an important determinant of fruit fly health, and domestication-related changes in the microbiome can have implications for the quality of fruit flies used in SIT programs [15].

Microbial communities are often highly abundant in insect digestive systems [16], especially bacteria [17,18]. Symbiotic bacteria can provide nutrition that contributes to insect host fitness [19,20], including amino acids [21] and essential vitamins [22], and nitrogen and carbon compounds [16,23]. Confirming the importance of the insect microbiome, elimination of resident bacteria can sharply reduce fruit fly fitness [24,25]. In tephritid fruit flies, changes in the gut microbiota through the developmental stages have been investigated in various species of *Bactrocera, Ceratitis, Anastrepha* and *Zeugodacus* including *B. tryoni* [11,26], *B. carambolae* [27], *B. dorsalis* [28], *B. latifrons* [29], *B. minax* [30,31], *C. capitata* [31], *A. ludens, A. obliqua, A. serpentina*, *A. striata* [32] and *Z. tau* [33].

*Bactrocera tryoni*, the Queensland fruit fly (‘Qfly’), is widespread in Eastern Australia where it is a significant pest in horticultural crops. SIT has been used to eradicate outbreaks in regions where permanent populations are not established, and to suppress populations in regions where permanent populations are established [34,35,36,37]. Given the importance of the microbiome to insect health, it is important to understand how the microbiome changes during the domestication of the Qfly. Some studies of the Qfly bacterial microbiome are available, characterising the bacteria associated with wild and domesticated larvae [11,26,38,39,40], pupae [11,26,41] and adult flies [11,26,42,43,44]. However, dynamics of the changes occurring in the Qfly microbiome in the transition from nature to artificial rearing have not been characterised.

In the present study, high-throughput Illumina sequencing was used to investigate bacterial diversity and abundance in the microbiome of the first five generations of Qfly, under artificial rearing conditions. Qfly colonies were established from wild material and were maintained through five generations of laboratory rearing on a gel-based artificial larval diet that is currently used to mass-rear flies for SIT programs. At each generation, the microbiome of larvae, pupae and adult males and females was assessed. Bacterial communities were significantly different between generations. Different bacterial genera were found to be highly abundant in all developmental stages of Qfly at G5 compared to the bacterial communities that were observed in the earlier generations (G1 to G4). This study greatly improves our understanding of changes in the Qfly microbiome during the early stages of domestication. This study has implications for factory-scale rearing, such as those required for SIT programs.

## 2. Materials and Methods

### 2.1. Colony Origins

Infested green apple *Malus pumila*, quince *Cydonia oblonga* and pomegranate *Punica granatum* were collected from various geographic locations in the Australian states of New South Wales (NSW) and Victoria (VIC) (Table 1). The Qfly infested fruit was collected from underneath trees, and most were over-ripe. After collection, all the fruit was stored in plastic bins (60 L, 447 × 236 × 663 mm, Award, Bunnings Warehouse, Australia) containing a 1 cm deep layer of fine vermiculite (Grade 1, Sage Horticultural, Hallam, VIC, Australia) in a controlled environment laboratory (25 ± 0.20 °C, 65 ± 3% RH and 11 h: 1 h: 11 h: 1 h light: dusk: dark: dawn photoperiod). A combined total of approximately 600 adult Qfly were obtained from this fruit. The emerged adult flies were provided hydrolysed yeast (MP Biomedicals, Cat. no 02103304) and commercial sucrose (CSR^®^ White Sugar, Maribyrnong, VIC, Australia) separately, and water through a moist sponge. Two populations, each of approximately 300 flies were maintained in mesh cages (Bugdorm 44545, 47.5 × 47.5 × 47.5 cm, MegaView Science Co., Ltd., Taichung, Taiwan) in a controlled environment room with the same conditions as described above and reared for five generations (G1 to G5).

### 2.2. Colony Maintenance

The Qfly colonies were reared on an artificial gel-based larval diet (Table 2) [11,45,46]. The gel-based diet was prepared as per [46], by mixing all the dry ingredients using a blender (Kenwood, Multipro FPM810 series, China) for 5 min. Water was mixed with agar and the solution was boiled. The dry mixture and the boiled agar were then mixed together. We transferred 150 mL of gel-based diet into larvae rearing containers (17.5 cm long, 12 cm wide and 4 cm deep) (Castaway Food Packaging, Arndell Park, NSW, Australia).

At each generation, eggs were collected using an oviposition device comprising a 300 mL semi-transparent white soft plastic bottle (low density polyethylene). The oviposition device had numerous ~1 mm holes through which females could oviposit, and each device contained 20 mL of water to maintain humidity. A few drops of natural apple juice were added to the water to attract the female flies and to encourage egg laying [47]. Eggs were collected from 14–16 day old flies between 9 am and 3 pm on a single day. The oviposition device was rinsed with distilled water to wash out the eggs. The eggs were then collected using a 50 mL falcon tube, and 250 µL of eggs-in-suspension were transferred to the larval diet using a 1000 µL pipette (approximately 3500 eggs, approximately 23 eggs per gram of diet) [46]. The larval rearing containers were then covered with plastic lids until the larvae reached their third instar and exited the diet to pupate. Subsequently, the rearing trays were placed in a container with a 1 cm layer of fine vermiculite on the bottom. The larvae exited the rearing container and pupated in the vermiculite. Pupae were collected by sifting them from the vermiculite. Approximately 600–800 pupae from each colony were placed in a mesh cage (Megaview Bugdorm 44,545, 47.5 × 47.5 × 47.5 cm, MegaView Science Co., Ltd., Talchung, Taiwan) for emergence. From each colony, third instar larvae (*n* = 6), 8 day old pupae (*n* = 6) and 15 day old sexually mature male (*n* = 6) and female flies (*n* = 6) were collected for sequencing at each generation from G1 to G5 (a total of 120 samples).

### 2.3. Sample Preparation

For sample processing, Qfly larvae, pupae and adult flies (male and female separately) were surface sterilized using 0.5% Tween 80 (Sigma–Aldrich, St. Louis, MO, USA, Cat. No. 9005656), 0.5% bleach (sodium hypochlorite) (Sigma–Aldrich, St. Louis, MO, USA, Cat. No.7681529) and 80% ethanol (Sigma–Aldrich, St. Louis, MO, USA, Cat. No. 65175) for 30 s, and rinsed 3 times in 1 M sterile phosphate-buffered saline (1× PBS) again for 30 s. The PBS from the 2nd and 3rd washes were kept and 100 µL spread plated on to five types of microbial growth medium (de Man, Rogosa and Sharpe Agar [MRSA], Tryptone Soya Agar [TSA], Macconkey Agar, Potato Dextrose Agar [PDA] and yeast-dextrose agar (YDA])) (Sigma–Aldrich, St. Louis, MO, USA) to check the performance of the sterilization method. All plates were incubated at 32 °C and 35 °C for 24 to 48 h [11,26,39,40]. Immediately after sterilisation, the gut of the adult flies was dissected using a stereomicroscope (Leica MZ6 stereo-microscope, Leica^®^, Wetzlar, Germany). Using sterile pestles, the larvae, the pupae, and the dissected guts from the adult flies were homogenised separately in a solution of Brain Heart Infusion (BHI) broth (Oxoid Ltd., Basingstoke, UK, Lot # 1656503) and 20% glycerol (Sigma–Aldrich, St. Louis, MO, USA, Lot # SHBG2711V) and each sample was stored in a separate cryovial tube (Simport Scientific, Saint-Mathieu-de-Beloeil, QC, Canada). All the samples were preserved at −80 °C. All procedures were completed in a sterile environment (Biological air clean bench, safe 2020 1.2, Thermo Scientific, Dreieich, Germany).

### 2.4. Microbiome Profiling

DNeasy Power Lyzer Power Soil Kit-100 (Qiagen, Hilden, Germany, Cat. no. 12855-100) was used for DNA extraction following the manufacturer’s protocol. Double nucleic acid (DNA) extracts were then quantified in the Invitrogen™ Qubit^®^ dsDNA High Sensitivity (HS) Assay Kit (Life Technologies, Eugene, OR, USA). Polymer chain reaction (PCR) amplification and sequencing were performed by the Australian Genome Research Facility, University of Adelaide, Plant Genomics Centre, Hartley Grove, URRBRAE, South Australia, 5064, Australia. For bacterial identification, the V1-V3 16S rRNA region was amplified using primers 27F (5′AGAGTTTGATCMTGGCTCAG-3′) and 519R (3′ GWATTACCGCGGCKGCTG-5′) [48] as used previously in [11,26,39]. Reactions contained 1X AmpliTaq Gold 360 mastermix (Life Technologies, Eugene, OR, USA), 0.20 µM of each forward and reverse primer and 25 µL DNA. PCR cycling conditions consisted of denaturation at 95 °C for 7 min, 35 cycles of 94 °C for 45 s, 50 °C for 60 s and 72 °C for 60 s, and a final extension of 72 °C for 7 min. A second PCR was used (PCR cycling conditions were the same as mentioned above) to adhere sequencing adaptors and indices to the amplicons. Primerstar max DNA Polymerase was used to generate a second PCR amplicon (Takara Bio inc., Shiga, Japan; Cat. No. #R045Q). The resulting amplicons were measured using a fluorimeter (Invitrogen Picogreen, Thermo Fisher Scientific, NSW, Australia) and normalised [49]. The normalised samples were pooled and quantified by qPCR prior to sequencing (Kapa qPCR Library Quantification kit, Roche, Basel, Switzerland). The resulting amplicon library was then sequenced on the Illumina MiSeq platform (San Diego, CA, USA) with 2 × 300 base pairs paired-end chemistry [50].

### 2.5. Sequence Data Processing

The Greenfield Hybrid Amplicon Pipeline (GHAP) was used to process bacterial 16 s rRNA amplicon sequences [51,52]. The Greenfield Hybrid Amplicon Pipeline (GHAP) is a publically available amplicon clustering and classification pipeline (https://doi.org/10.4225/08/59f98560eba25, published on 1 November 2017) [51] built around tools from USEARCH [53] and the Ribosomal Database Project (RDP) [54], combined with locally written tools for demultiplexing, trimming and generating OTU (Operational Taxonomic Unit) tables. This hybrid pipeline produces a table of taxonomically assigned OTUs and their associated read counts across all samples. First, the amplicon reads were demultiplexed and trimmed, and the read pairs were then merged (using *fastq_mergepairs*) and de-replicated (using *fastx_uniques*). The merged reads were then trimmed again and clustered at 97% similarity (using *cluster_otus*) to generate OTUs. Representative sequences from each OTU were then classified both by finding their closest match in a set of reference 16S sequences (using *usearch_global*), and by using the RDP naïve Bayesian classifier. The pipeline mapped the merged reads back onto the classified OTU sequences to get accurate read counts for each OTU/sample pairing and generated an OTU table complete with taxonomic classifications and species assignments. The OTU table was then summarised over all taxonomic levels, combining the counts for identified taxa across all OTUs. The pipeline finally classified all the merged reads using the RDP classifier, regardless of whether they were assigned to an OTU. This last step was to provide confidence in the clustering and OTU formation steps by providing an independent view of the community structure.

All OTUs assigned to ‘mitochondria’ at the Order level, were removed from the dataset before downstream processing. The biome table, described above, was rarefied to 5000 reads per sample, repeated 50 times and the counts were averaged to obtain a representative rarefaction in order to maintain equal sequence depth among all samples. This was achieved using an in-house Python script. Those samples with <5000 reads were excluded. The data were then normalised as the percentage of relative abundance, and are henceforth referred to as the OTU table (Appendix A). All the measurements of bacterial relative abundance, at different developmental stages, and between generations in colonies, reared on different diets, were plotted in Prism 8 (version 8.0.1(145), GraphPad software, Inc. San Diego, CA, USA) as used previously in Majumder et al. (2019, 2020a,b,c) [11,26,40,44]. The Illumina sequence data were deposited into and made publicly available in the NCBI database under Bioproject PRJNA717989.

### 2.6. Microbiome Analysis

The OTU table was imported into Primer-E v7 for analysis as described in Clarke and Ainsworth, 1993; Sutcliffe et al., 2017, Majumder et al., 2019, 2020a, b, c [11,26,34,40,44,52]. The DIVERSE function was used to generate univariate biodiversity metrics, species richness and the Shannon and Simpson biodiversity indices. Statistical differences between these metrics were assessed in the JMP Statistical Software Version 10.0.0 (SAS Institute, Cary, NC, USA) using one-way analysis of variance (ANOVA) and Tukey–Kramer post hoc analysis. The Operational Taxonomic Unit (OTU) table was first log transformed using Primer-E v7 to observe the taxonomic compositional changes for the bacterial communities. A Bray–Curtis similarity matrix was derived from this transformed data and a permutation analysis of variance (PERMANOVA) pairwise comparison was conducted to compare all community samples. A *p* value of < 0.05 was considered statistically significant. Further, ordination plots of these communities were visualised using principal coordinates analysis (PCoA) in Primer-E. We performed ANOVA and post-hoc Tukey–Kramer tests to determine whether significant differences occurred in the relative abundance of bacterial communities at all developmental stages, as well as generations, of the Qfly. 

## 3. Results

### 3.1. 16s rRNA Sequence Reads and OTUs

We sequenced the bacterial microbiome of 120 Qfly samples from five consecutive generations (G1 to G5) reared on a gel-based larval diet. Among them, 115 were retained after quality control and rarefaction at 5000 reads per sample (five samples were removed). After rarefaction and quality control, a total of 275 bacterial OTUs (operational taxonomic units) were detected across the 115 samples (Appendix A).

### 3.2. Bacterial Alpha and Beta Diversity during Qfly Domestication

Bacterial alpha biodiversity metrics, including the Shannon biodiversity and species richness indices were compared between generations (Figure 1) as well as the developmental stages of different generations (Figure 2 and Figure 3). The bacterial species richness indices were significantly different in G1 compared to other generations (*p* < 0.05) (Figure 1a). However, no significant difference was found in the Shannon index between the generations (Figure 1B).

This appears to be driven by a significantly higher level of species richness in the larvae from G1 and G5, when compared with larvae from G2, G3 and G4 respectively (Figure 2A).

In contrast, the Shannon index was significantly higher in larvae from G1 compared to other generations (Figure 3A). None of the bacterial alpha diversity metrics showed significant differences in the pupal microbiome between generations (G1 to G5) (Figure 2B and Figure 3B). The Shannon biodiversity and species richness indices were significantly different both in the adult male and female microbiome collected from G1 and G2 compared with G3, G4 and G5 respectively (Figure 2C,D and Figure 3C,D).

Beta diversity of the bacterial communities at each Qfly stage was assessed by PERMANOVA analysis (pair-wise test with 999 permutations) based on Bray–Curtis similarities (Appendix A). Additionally, a principal coordinate analysis (PCoA) of this Bray–Curtis similarity matrix was used to visualize variation among host microbial communities (Figure 4 and Figure 5). The PERMANOVA demonstrated a significant difference in bacterial communities of Qfly among generations (PERMANOVA < 0.05, Appendix A).

However, the PCoA ordination plot showed that the microbiome from G1 and G2 were positioned in the same cluster and separately located from other generations (G2 to G5) (Figure 4). We observed no differences between the male and female microbiome across all generations (PERMANOVA test, *p* = 1.57, Appendix A). In addition, all developmental stages in G5 were found separately clustered from other generations in the PCoA ordination pot (Figure 5A–D).

### 3.3. Bacterial Communities Associated with Domestication of the Qfly

After a comprehensive taxonomic analysis, a total of 6 phyla, 38 families and 62 genera were observed in the dataset. The relative abundance of the bacterial community members in the microbiome of each generation, as well as each developmental stage, including larval, pupal and adults (both male and female) stage, was analysed.

At the phylum level, the most abundant taxa in the Qfly microbiome were Proteobacteria (51%), followed by Firmicutes (29%) and Actinobacteria (4%) (Appendix A). Unassigned bacteria were found with an average relative abundance of 16% in the Qfly microbiome. Among all generations, the relative abundance of the Proteobacteria was highest at 93% in G5, and lowest in G3 (31%). In the larval stages, compared to other generations, the relative abundance of the Proteobacteria was highest (99%) in G5. Actinobacteria was the lowest (1%). Similar results were observed in the pupal stages. In the adult microbiome, Proteobacteria were observed at 96% in females and 90% in males from G5. Only Proteobacteria and Firmicutes were found both in adult male and female Qfly microbiomes across all generations from G1 to G5.

At the family level, the most prevalent taxa were Enterobacteriaceae, which represented an average relative abundance of 41% in all generations (Appendix A). Although, they were observed with the highest relative abundance in G5 (61%). Conversely, the presence of Staphylococcaceae in G1 was highest with a relative abundance of 26% and was completely absent in G5. In the larvae stages, the Enterobacteriaceae was present in all generations, but highest relative abundance was in G5 at 99.5%. In the pupal stages, Burkholderiaceae and Enterobacteriaceae were both only present in G5 (75% and 10%, respectively) compared to other generations (G1 to G4). Conversely, Enterobacteriaceae and Enterococcaceae were abundant in adult males and females in all generations, however, the Aeromonadaceae was observed to be highly abundant in both males and females in G5 (38% and 13%, respectively).

At the genus level, bacterial taxa with the greatest relative abundance across the dataset were unassigned bacteria (21%), *Staphylococcus* (17%), *Citrobacter* (9%), *Enterobacter* (8%), *Klebsiella* (8%), *Vagococcus* (7%) *Morganella* (6%), *Providencia* (4%), *Burkholderia* (4%), *Lactococcus* (4%), *Aeromonas* (3%), *Kluyvera* (2%) and *Arthrobacter* (1%) (Figure 6). *Staphylococcus* was found in low abundance in G1 (6%) and gradually increased to >30% in G2 to G4, however, a sharp decrease (5%) was observed in G5. Furthermore, in G5, *Morganella* (26%), *Providencia* (10%), and *Burkholderia* (19%) were found to be highly abundant compared to other generations (G1 to G4) (Figure 6).

The abundant taxa were identified across the Qfly life stages through all generations. In the larval stage, unassigned bacteria were found highly abundant in all generations but were absent in G5 (Figure 7A). However, *Morganella* was highly abundant in larvae from the colony at G5 (96%) but had low abundance in the larvae from other generations. Similarly, *Enterobacter* (49%), *Kluyvera* (8%) and *Citrobacter* (5%) were abundant only in the larvae from G1 (Figure 7A).

In pupae, *Staphylococcus* was highly abundant (>68%) in G1 to G4 but was only found at very low abundance (1%) in G5. The opposite was found for *Burkholderia*, which was highly abundant in the pupae (75%) of G5 but absent in other generations (Figure 7B). In adult males, bacterial genera of *Providencia* and *Aeromonas* (both commonly pathogenic) were highly abundant in G5 (25% and 38% respectively) (Figure 7C). In contrast, *Klebsiella* was found with a high relative abundance of 53% in G1 but was absent in G4 and G5. In adult females, *Kluyvera* and *Aeromonas* were particularly common in G5 but were not detected in other generations (G1 to G4) (Figure 7D).

## 4. Discussion

This study provides a detailed analysis of how the domestication process affects the microbiome of Qfly across the first five generations of rearing under artificial conditions. Using high-throughput Illumina sequencing methods, we were able to address meaningful questions regarding the dynamics of the microbiome in Qfly during domestication in larvae, pupae, and adult males and females. Our findings suggest that the domestication process strongly modulated the microbial community structure (beta diversity) across all developmental stages but did not affect total biodiversity (as assessed by alpha diversity metrics: species richness and the Shannon diversity index). Profiling of the bacterial elements present in the domesticated Qfly microbiome circumvents the well-known difficulties in isolating microbes through culture-dependent methods. The bacterial communities identified across generations were novel, lacking closely-related described culture representatives and were easy to demonstrate between generations. This approach will enable us to interrogate how the Qfly microbiota changes through domestication in mass rearing and will be helpful in assessing biodiversity independent of culturing and manipulating the bacteria during mass rearing.

Beta diversity analysis found substantial and significant shifts in the bacterial communities from G1 to G5. The bacterial communities at G5 were found to cluster separately from other generations. Also, similar results were observed when compared between each developmental stage across the generations. This finding strongly indicates an effect of domestication on microbial beta diversity under artificial rearing. Previous studies commonly found the bacterial phyla Proteobacteria and Firmicutes in Qfly larvae and adults from domesticated colonies reared on artificial larval diets [11,38,44]. Furthermore, Proteobacteria and Firmicutes have also commonly been observed in other fruit flies including *B. neohumeralis*, *B. jarvisi*, *B. cacuminata, Zeugodacus tau, Anastrepha ludens, A. obliqua, A. serpentina*, *A. striata* and *C. capitata* [31,32,33,42]. However, at the phylum level, microbial community structure showed only limited variation across generations and developmental stages. Variation was observed mostly at the level of the family and/or genus. Despite the substantial changes in the overall microbiome across the generations, some bacterial families and genera showed consistent trends when comparing between the generations. *Citrobacter, Enterobacter* and *Lactococcus* from the family Enterobacteriaceae were found across all generations. *Citrobacter* has often been reported in other fruit flies, including *C. capitata* [55,56], *B. tau* [57], *B. zonata* [58], *B. oleae* [59] and *A. ludens* [60]. Specifically, G1 to G4 were associated with an increased relative abundance of the Staphylococcaceae genera *Staphylococcus* while this taxon was almost non-existent at G5. Previous research on the Qfly microbiome across the developmental stages of Qfly at G5 were consistent with our findings [11]. Conversely, G5 exhibited an increased abundance of Enterobacteriaceae, *Morganella, Providencia* and *Burkholderia*. Surprisingly, these bacteria were found only at very low abundance in G1 and G2. In previous studies, Enterobacteriaceae genera *Morganella* and *Providencia* were also observed at only low abundance in both wild and domesticated Qfly reared on carrot diet [11,26,39].

The present study observed that in artificial rearing, bacteria with high relative abundance in early generations (for example, *Staphylococcus*, *Enterobacter, Klebsiella, Vagococcus*) are of greatly reduced abundance or are replaced by other bacteria at G5 (Figure 6). The same trend was also observed at each development stage between different generations where bacterial abundance sharply changed (Figure 7). For example, in larval and pupal stages at G1 to G4, Unassigned bacteria and *Staphylococcus* were highly abundant. However, at G5 Unassigned bacteria and *Staphylococcus* were replaced by *Morganella* and *Burkholderia*. Compared to larval and pupal stages, we observed less variation in bacterial abundance between generations at the adult stage. Diet could be a key factor responsible for these microbial changes [11], as is the case in other insects e.g., cotton bollworm *Helicoverpa armigera* [61], ground-dwelling beetles (Coleoptera) [62], gypsy moth, *Lymantria dispar* L. [17] and *Drosophila* [63]. Some microbes may be acquired through diet, and diet is a major exogenous factor that can directly influence the composition of insect gut microbial communities and their metabolic capabilities [39,64,65,66,67]. Additionally, variation in the diet’s nutritional composition (protein, carbohydrate and lipid) can influence both the gut microbiome biodiversity and community structure [17,44,68].

## 5. Conclusions

The present study identified and characterised the microbial communities present in each developmental stage of the Qfly across multiple generations of domestication from G1 to G5. Overall, we found that the microbial community structure changed significantly across developmental stages as well as between generations. This knowledge has applied value, providing guidance for potential interventions that might maintain select components of the microbiome through domestication to improve the physiology and behaviour of the larvae and adults, thereby improving the quality of mass reared Qfly for SIT.

## Figures and Tables

**Figure 1 microorganisms-10-00291-f001:**
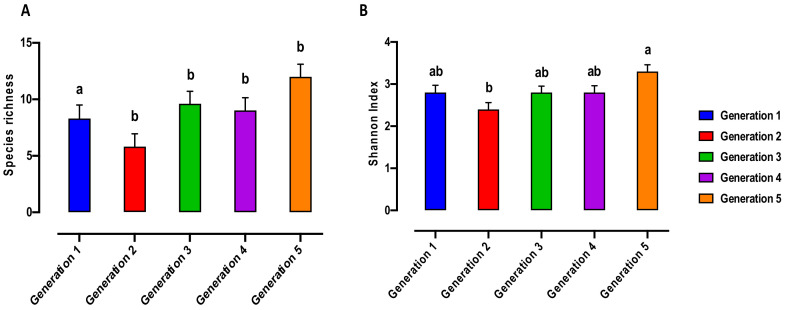
Alpha diversity of the bacterial microbiome of the Qfly from G1 to G5 includes (**A**) Species richness and (**B**) Shannon indices. Different letters indicate significant Tukey’s post hoc comparisons (*p* < 0.05).

**Figure 2 microorganisms-10-00291-f002:**
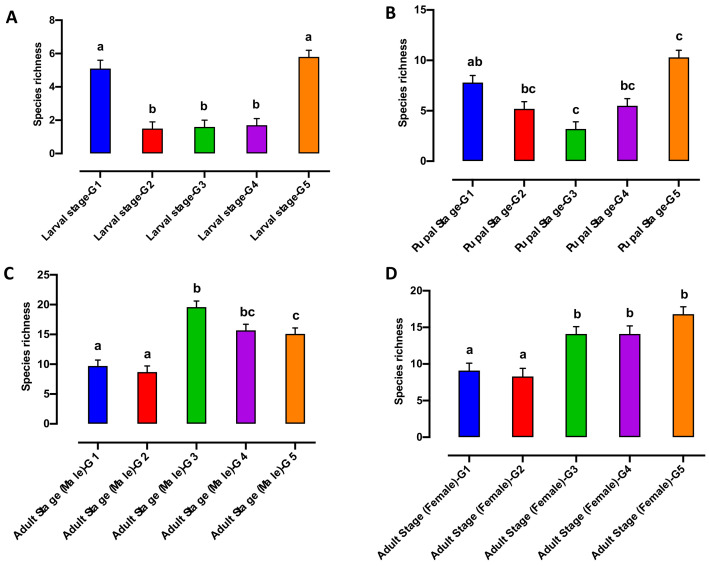
Species richness of the bacterial microbiome of the Qfly developmental stages from G1 to G5 (**A**) Larval stage; (**B**) Pupal stage; (**C**) Adult stage (Male) and (**D**) Adult stage (Female). Different letters indicate significant Tukey’s post hoc comparisons (*p* < 0.05).

**Figure 3 microorganisms-10-00291-f003:**
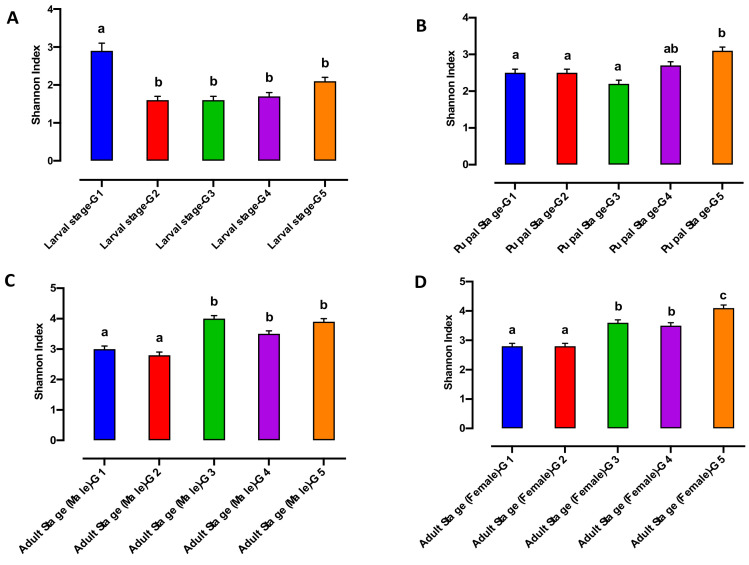
Shannon index of the bacterial microbiome of the Qfly developmental stages from G1 to G5 includes (**A**) Larval stage; (**B**) Pupal stage; (**C**) Adult stage (Male) and (**D**) Adult stage (Female). Different letters indicate significant Tukey’s post hoc comparisons (*p* < 0.05).

**Figure 4 microorganisms-10-00291-f004:**
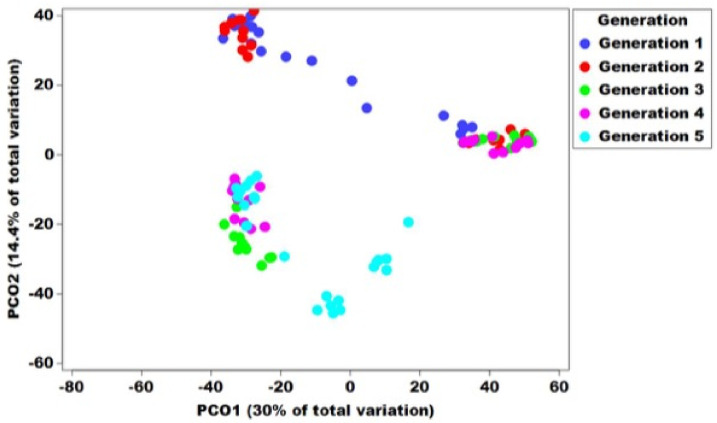
Principal co-ordinate analysis of five consecutive generations of the Qfly during the domestication process from Generation 1 to Generation 5 reared on a gel-based larval diet. Different colours indicate the microbial communities in the different generations of the Qfly.

**Figure 5 microorganisms-10-00291-f005:**
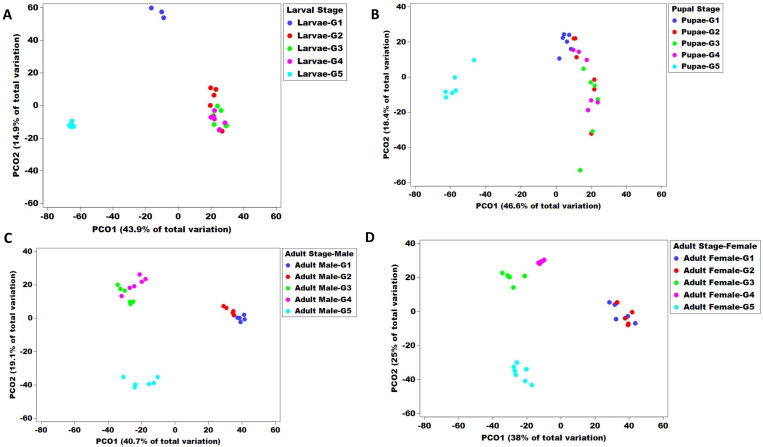
Principal co-ordinate analysis of five consecutive generations of the Qfly across all developmental stages during the domestication process from Generation 1 to Generation 5 reared on gel-based larval diet. (**A**) larvae; (**B**) Pupae; (**C**) Adult Male and (**D**) Adult Female. Different colours indicate the microbial communities in the different generations of the Qfly.

**Figure 6 microorganisms-10-00291-f006:**
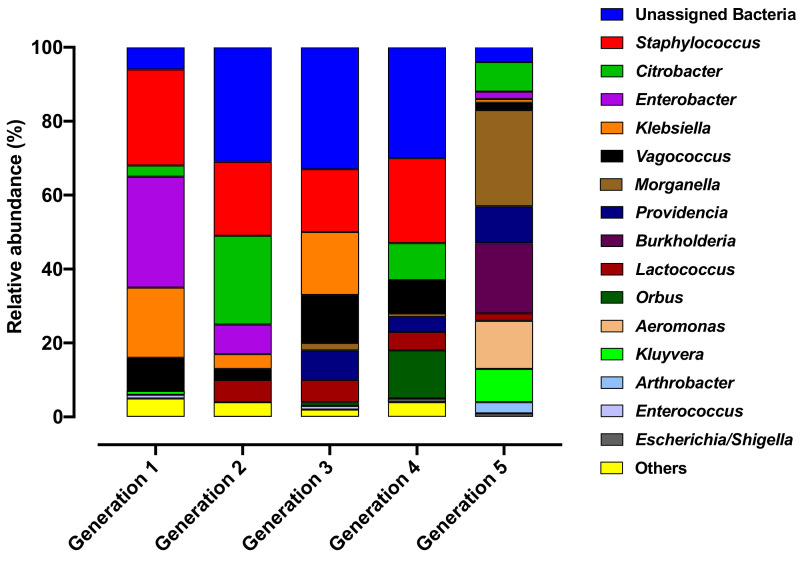
Comparative average relative abundance of the bacterial genera present in five consecutive generations of the Qfly during the domestication process from Generation 1 to Generation 5 reared on a gel-based larval diet.

**Figure 7 microorganisms-10-00291-f007:**
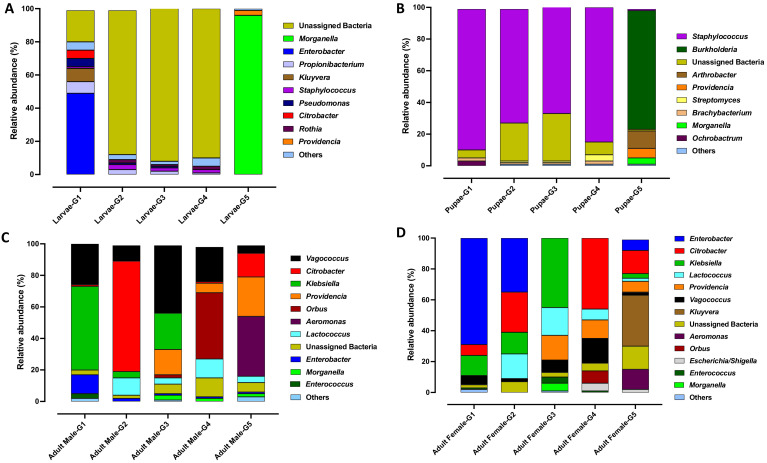
Comparative average relative abundance of the bacterial genera present in five consecutive generations of the Qfly across all developmental stages during the domestication process from Generation 1 to Generation 5 reared on a gel-based larval diet. (**A**) Larvae; (**B**) Pupae; (**C**) Adult Male and (**D**) Adult Female.

**Table 1 microorganisms-10-00291-t001:** Fruit types and origin for wild *Bactrocera tryoni* larvae collection. A total of six replicate larvae, and fruit flesh samples were collected from each fruit origin.

Geographic Location of Collection	Fruit Source and Number of Fruit Collected	Collection Date
Coomealla, NSW GPS: Lat 34°5′50.97′′, Long. 142°3′7.21′′	Pomegranate (*Punica granatum)* 37 pieces	5 May 2017
St. Germains, between Tatura and Echuca, Victoria GPS: Lat 36°10′48.86′′, Long. 145°8′50.74′′	Green Apple *(Malus pumila)* 41 pieces	5 May 2017
Downer Road, between Tatura and Toolamba, Victoria GPS: Lat 26°38′34.92′′, Long. 152°56′22.99′′	Quince *(Cydonia oblonga)* 52 pieces	5 May 2017

**Table 2 microorganisms-10-00291-t002:** Gel-based larval diet recipe.

Ingredients	1 kg Diet Preparation	Company Name and Catalogue Number
Brewer’s Yeast (g)	204	Lallemand Australia Pty Ltd., Edwardstown, SA, Australia
Sugar (g)	121.8	MP Biomedicals LLC, France (Cat. n^o^02902978)
Agar (g)	10	Sigma–Aldrich^®^ St. Louis, MO, USA
Citric Acid (g)	23.1	Sigma–Aldrich^®^, St. Louis, MO, USA
Nipagin (g)	2	Southern Biological, Knoxfield, VIC, Australia (Cat n^o^ MC11.2)
Sodium benzoate (g)	2	Sigma–Aldrich^®^ St. Louis, MO, USA
Wheat Germ Oil (mL)	2	Melrose Laboratories Pty Ltd., Australia
Water (mL)	1000	Milli-Q water

## Data Availability

The Illumina sequence data were deposited and publicly available in NCBI database under Bioproject PRJNA717989.

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
