# Peer review of "Dynamics of the Queensland Fruit Fly Microbiome through the Transition from Nature to an Established Laboratory Colony"

_microorganisms, 2022, doi:10.3390/microorganisms10020291_

Round 1
Reviewer 1 Report
This paper describes changes in the microbiome of the different developmental stages of Bactrocera tryoni maintained for five generations under artificial rearing. The article is well written and deepens the understanding of the relatively poorly researched issue, i.e. the changes in the structure of the microbial community of insects subjected to domestication processes. In my opinion, the experimental design is correct, and the experimental part is mostly well conducted. However, I have some comments and suggestions.
Although the work is generally written grammatically and linguistically correctly, I suggest checking it again in this respect, as there are still fragments of sentences that need improvement. A few examples of texts that require editing:
Line 107: "compare with G3, G4 and G5" - It should be "compared with"
Line 151 - "with highest relative abundance in G5" - It should be "with the highest"
Line 208 - the word "also" is duplicated in the sentence
Line 241 - "ground dwelling beetles" - It should be "ground-dwelling beetles"
Additionally, I noticed an inconsistency in the use of the English versions. According to the Instructions for Authors, American English or UK English are fine so long as there is consistency. However, in the paper you can find both the words "sterilisation" (line 34) and "sterilization (line 304). Likewise, there are the words "normalised" and "normalized" (lines 362 and 331) as well as "analysed "and "analyzed" (lines 393 and 134). Please use the chosen English version consistently.
I also have a few comments about the figures. Axis labels and some legends of Figure 5A are almost unreadable due to their too small size, especially when printed. I suggest enlarging them.
Figures 7 A-D are mutually difficult to compare. In the Figures 7 A-D legends, individual genera should be arranged as shown in Figure 6. Currently, they are ordered inversely, i.e. the lowest bacteria in the figure bars are the highest in the legend, which makes it difficult to analyze the figures.
In addition, I suggest standardizing the ordering of the genera in Figures 7 A-D. This is not currently the case, for example, Morganella and Enterobacter are side by side at the top or bottom of the figures, while in Figure 7D they are on opposite parts of the figure bars. Likewise, unassigned bacteria represent the lowest level of the bars in Figure 7A, while in the rest of the figures they are found elsewhere (otherwise, in Figure 7D, unassigned bacteria are shown in a different color). Unifying the order of appearance and colors of individual groups of bacteria in Figures 7A-7D will definitely improve their readability.
I have a few more comments about the results presented in Figure 7A. The authors do not provide an explanation why unassigned bacteria were found to be so highly abundant in the larval stage. Is it typical for the fruit fly microbiome? Maybe a different, more variable region of the 16S gene should be amplified and sequenced using the Illumina MiSeq or the workflow regarding the bioinformatics analysis should be changed? I believe the authors should explain this issue in the Discussion.
Reviewer 2 Report
This manuscript describes changes in the microbial communities associated with the Qfly as it undergoes adaptation to laboratory rearing. This reviewer is not surprised that changes occur, and finds the manuscript for the most part clearly written. Although these results are interesting, they are not compelling in the apparent absence of biological replication.
A few points deserve attention:
In the abstract, line 22 is hard to understand: "However, communities converged at Phyla to family taxonomic levels. "
I think the concept is better described in lines 74-76 in the introduction.
The manuscript could be improved by adding a more robust description of Shannon index and its implications for this study.
A list of the bacteria identified in this study would be a welcome addition for those interested in the microbiology.
Minor point: letters designating statistical differences are not described in the figure legends.
In Fig 2, I find the decrease in generations 2, 3 and 4 biologically surprising, relative to generation 5. Why the drop and recovery? Are some bacterial species entering and then recovering from a persister state?
The transitions in Figure 6 need more discussion. Has there been a biological replicate that supports these observations?
Reviewer 3 Report
The manuscript entitled "Dynamics of the Queensland fruit fly microbiome through the transition from nature to established laboratory colony" describes the microbiota changes through generations after establishment of a field colony in laboratory conditions. The methods are well adequate and the results well described. I recommend publication at the current form.
Round 2
Reviewer 2 Report
These authors have appropriately addressed reviewer comments, and I have no further concerns.